# Single-cell analysis based dissection of clonality in myelofibrosis

Elena Mylonas [1,11], Kenichi Yoshida [2,11], Mareike Frick[1,11], Kaja Hoyer[1], Friederike Christen[1], Jaspal Kaeda[1], Matthias Obenaus[1], Daniel Noerenberg[1], Cornelius Hennch [1], Willy Chan[1], Yotaro Ochi[2,3], Yuichi Shiraishi[4], Yusuke Shiozawa[2], Thorsten Zenz[5], Christopher C. Oakes[6], Birgit Sawitzki[7], Michaela Schwarz[1], Lars Bullinger[1,8], Philipp le Coutre[1], Matthew J.J. Rose-Zerilli[9], Seishi Ogawa[2,3,10] & Frederik Damm[1,8]*

Cancer development is an evolutionary genomic process with parallels to Darwinian selection. It requires acquisition of multiple somatic mutations that collectively cause a malignant phenotype and continuous clonal evolution is often linked to tumor progression. Here, we show the clonal evolution structure in 15 myelofibrosis (MF) patients while receiving treatment with JAK inhibitors (mean follow-up 3.9 years). Whole-exome sequencing at multiple time points reveal acquisition of somatic mutations and copy number aberrations over time. While JAK inhibition therapy does not seem to create a clear evolutionary bottleneck, we observe a more complex clonal architecture over time, and appearance of unrelated clones. Disease progression associates with increased genetic heterogeneity and gain of RAS/RTK pathway mutations. Clonal diversity results in clone-specific expansion within different myeloid cell lineages. Single-cell genotyping of circulating CD34 + progenitor cells allows the reconstruction of MF phylogeny demonstrating loss of heterozygosity and parallel evolution as recurrent events.

[1] Charité—Universitätsmedizin Berlin, Corporate Member of Freie Universität Berlin, Humboldt-Universität zu Berlin, and Berlin Institute of Health, Department of Hematology, Oncology, and Tumor Immunology, Berlin, Germany. [2] Department of Pathology and Tumor Biology, Graduate School of Medicine, Kyoto University, Kyoto, Japan. [3] Institute for the Advanced Study of Human Biology (WPI-ASHBi), Kyoto University, Kyoto, Japan. [4] Laboratory of Sequence Analysis, Human Genome Center, Institute of Medical Science, The University of Tokyo, Tokyo, Japan. [5] Department of Medical Oncology and Hematology, University Hospital Zurich / University of Zurich, Zurich, Switzerland. [6] Division of Hematology, Department of Internal Medicine, The Ohio State University, Columbus, OH, USA. [7] Charité—Universitätsmedizin Berlin, Corporate Member of Freie Universität Berlin, Humboldt-Universität zu Berlin, and Berlin Institute of Health, Institute for Medical Immunology, Berlin, Germany. [8] German Cancer Consortium (DKTK) and German Cancer Research Center (DKFZ), Heidelberg, Germany. [9] Cancer Sciences, Faculty of Medicine, University of Southampton, Southampton, UK. [10] Department of Medicine, Centre for Haematology and Regenerative Medicine, Karolinska Institute, Stockholm, Sweden. [11]These authors contributed equally: Elena Mylonas, Kenichi Yoshida, Mareike Frick *email: frederik.damm@charite.de

Cancer conforms a group of diseases that arise from a single-cell, characterized by uncontrolled proliferation, resistance to apoptosis, independence from environmental control signals and nutritional restrictions, and genetic instability[1]. Tumor cells that clonally expand acquire different mutations resulting in the development of genetically heterogeneous subclones, which will be subjected to selection[2–4]. During cell proliferation, mutations can stochastically be acquired and lost but their maintenance or fixation in the tumor population will depend on cellular properties in the context of the environment and the disease phase, such as proliferative advantage during the onset of carcinogenesis or chemotherapy resistance during treatment. Chemotherapy treatment can be seen as bottleneck, having a direct impact on tumor architecture and clonal heterogeneity, since it can open space for the outgrowth of chemoresistant clones, thereby resulting in treatment failure and relapse[5–8]. In addition, chemotherapy can induce DNA damage and thus foster the appearance of novel mutations[9].

BCR-ABL-negative myeloproliferative neoplasms (MPN) are a heterogeneous group of malignant diseases mainly consisting of polycythemia vera (PV), essential thrombocythemia (ET) and primary myelofibrosis (PMF). While patients with PV and ET show often a relatively mild clinical course and only sometimes require chemotherapeutic intervention, they can progress to secondary myelofibrosis (post-ET/PV-MF). Myelofibrosis, primary or secondary, is a life-threatening condition characterized by progressive deterioration of the bone marrow, enhanced circulation of hematopoietic progenitor cells and development of extramedullary hematopoiesis. Ten to 20% of MF patients progress to acute myeloid leukemia (AML)[10,11]. Constitutive activation of JAK2 signaling through somatic mutations affecting JAK2, MPL, or CALR is a hallmark of MPN pathogenesis and represents a therapeutic target[12–15]. Large-scale sequencing studies have unraveled the mutational landscape of MPN, demonstrating clonal heterogeneity and importance of genetically defined subgroups in disease prognosis and progression[16–19]. Importantly, the order in which JAK2 and TET2 mutations were acquired influenced the response to targeted therapy, and clonal evolution in MPN patients[16]. JAK inhibitors have been shown to improve clinical symptoms and are nowadays standard of care for intermediate/high-risk MF patients[20,21]. However, JAK2/CALR mutant allele burden is reduced only modestly during treatment in most cases. In addition, while clonal evolution has been reported in up to one third of MF patients during ruxolitinib treatment[22], investigation was limited to a set of selected genes and thus genome-wide changes remain poorly understood.

In order to investigate the genetics of MF progression and its molecular drivers during JAK inhibition therapy, we perform in-depth genetic studies on longitudinal blood samples from 15 MF patients covering a disease span of 3 to 5 years after initiation of ruxolitinib. Whole-exome sequencing (WES) is used at several time points to study the mutational diversification and clonal evolution during treatment. Single-cell genotyping of circulating CD34 + progenitor cells allows us to reconstruct the phylogeny and subclonal composition of MF. Collectively these data recapitulate the mutational history of the disease, the initiating/predisposing events and its evolution. Albeit the chronic nature of MF and apparent stability of mutations over time, we detect clonal composition changes, reversion and parallel evolution.

## Results

**Whole-exome sequencing of samples during ruxolitinib therapy.** Sequential samples from 15 MF patients (PMF $n = 8$; post-ET/PV-MF $n = 7$; median age 66 years) accounting for a total of 42 time points representing 58.5 years of ruxolitinib treatment (mean follow-up time: 3.9 years/patient) were investigated by WES as depicted in the CONSORT diagram (Supplementary Fig. 1). The first sample was collected at initiation of treatment and the last time point was collected at last visit ($n = 10$), leukemic transformation ($n = 3$), or before death without leukemic transformation ($n = 2$). Two patients achieved a molecular remission and were clinically stable for the period of the study. Clinical characteristics of each patient are summarized in Table 1.

WES with a median depth of $171 \times$ (range of $126–232 \times$, Supplementary Data 1) was performed on 2 to 4 time points per patient. One follow-up exome was performed for four patients, nine patients had two follow-up and a single-patient three follow-up exomes. In vitro expanded T-cells were used as germline control. Consistent with previous genetic characterizations of MF[13], WES at initiation of ruxolitinib treatment (= baseline

**Table 1 Baseline characteristics of the 15 investigated myelofibrosis patients.**

| Patient | Age at baseline | Sex | Diagnosis | Disease progression | Karyotype at baseline WES | Response to ruxolitinib | alive/dead | Cause of death |
|---|---|---|---|---|---|---|---|---|
| MPN01 | 73 | Male | PMF | No | 46, XY | Durable response | Alive | n.a. |
| MPN02 | 64 | Male | PMF | Yes | 46,XY,del(11)(q13-14q23) | Progression to AML | Dead | AML |
| MPN03 | 65 | Female | Post-PV-MF | No | 46,XX | Durable response | Alive | n.a. |
| MPN04 | 70 | Female | Post-ET-MF | Yes | 46, XX, del (5)(q23q32) | Progression to AML | Dead | AML |
| MPN05 | 70 | Male | PMF | No | 46, XY | Durable response | Alive | n.a. |
| MPN06 | 75 | Female | PMF | No | 46,XX | Durable response | Alive | n.a. |
| MPN07 | 59 | Female | Post-ET-MF | No | 46,XX | Durable response | Alive | n.a. |
| MPN08 | 72 | Male | PMF | No | 46,XY | Durable response | Alive | n.a. |
| MPN09 | 48 | Female | Post-PV-MF | No | 46,XX | Molecular remission | Alive | n.a. |
| MPN10 | 76 | Female | PMF | No | 46,XX | Durable response | Dead | Heart failure |
| MPN11 | 75 | Female | Post-PV-MF | No | 46,XX | Durable response | Dead | Multi-organ failure following complicated colon cancer operation |
| MPN16 | 67 | Male | PMF | No | 46,XY | Durable response | Alive | n.a. |
| MPN17 | 47 | Female | Post-PV-MF | No | 46,XX | Durable response | Alive | n.a. |
| MPN18 | 64 | Male | PMF | Yes | 46,XY | Disease acceleration | Dead | Sepsis in disease progression |
| MPN19 | 54 | Male | Post-PV-MF | No | 46,XY | Molecular remission | Alive | n.a. |

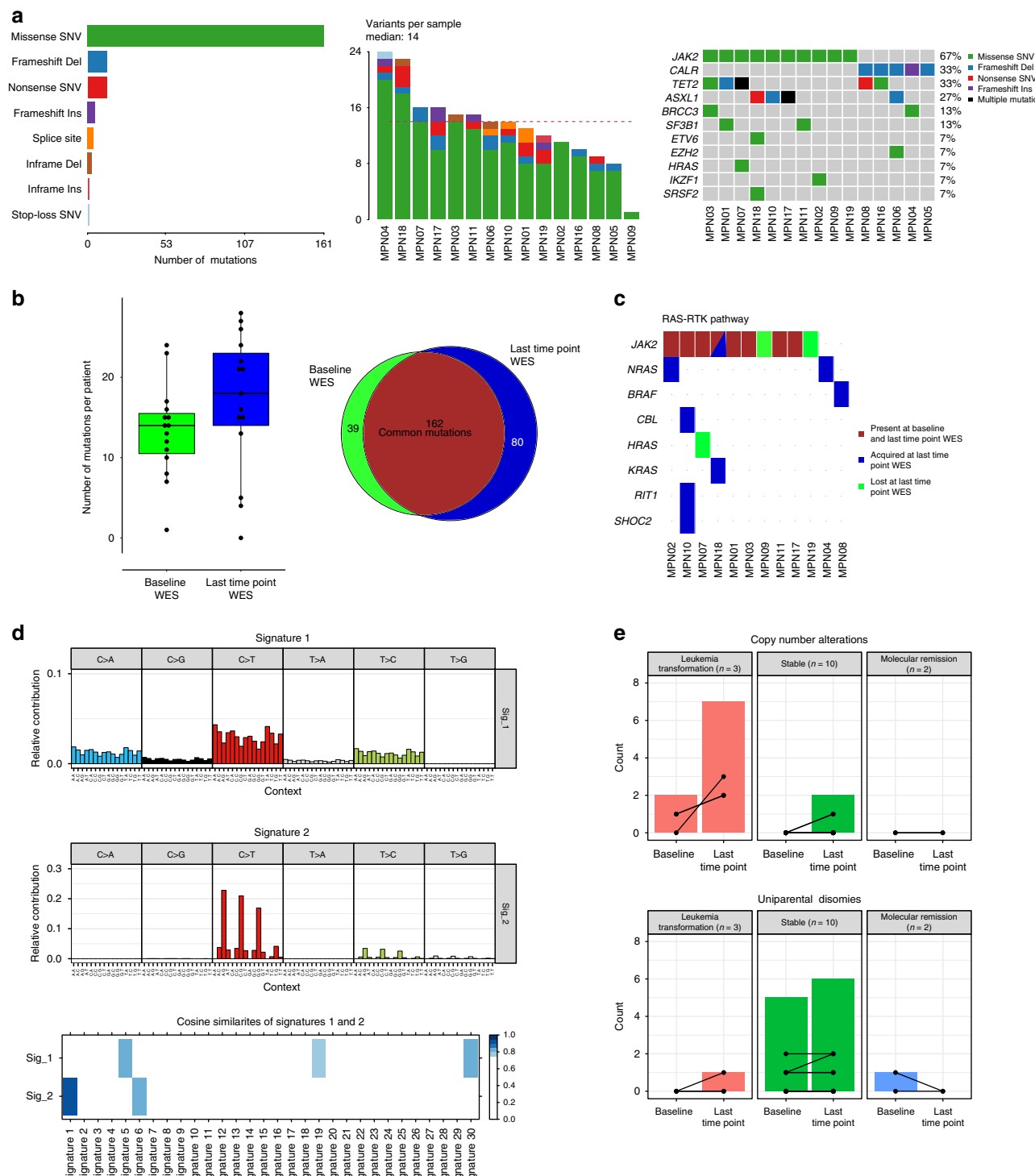

**Fig. 1 Mutational landscape in MF. a** Type, number, and most frequent mutations in 15 MF patients. **b** Number of gained and lost mutations comparing baseline and last time point WES. **c** Mutations affecting genes of the RAS-RTK pathways. **d** Mutation signatures analysis identified two main signatures at baseline and last time point WES and their respective cosine similarities with established COSMIC signatures. **e** Number of CNA and CN-LOH per patient. Source data are provided as a Source Data file.

WES) identified a median of 14 non-silent somatic mutations (range 3 to 24). Across all patients, we detected 179 somatic single-nucleotide variants (SNVs) and 22 somatic insertions and deletions (indels) in a total of 174 mutated genes (Fig. 1a and Supplementary Data 1).

In our cohort, ten patients had a *JAK2* V617F and five patients a *CALR* mutation (four Type-A, one Type-B) as disease-defining

alterations. At baseline, the most frequently mutated genes detected by WES were *TET2* in 33% (5/15) and *ASXL1* in 27% (4/15), followed by *SF3B1* and *BRCC3* in 13% (2/15) of patients each. While mutations in *BRCC3*, a metalloprotease implicated in DNA repair, have been recurrently reported in myelodysplastic syndromes (MDS) and AML[23,24], they have not previously been reported in MF.

When comparing mutations between first and last investigated time points, the majority of baseline mutations (162/201 = 81%) could be detected also at a later disease stage (Fig. 1b). A total of 39 mutations were lost and 80 new mutations were detected at the last time point, indicating an evolutionary process. All 15 patients showed at least one gained and/or lost mutation in sequential samples (Supplementary Fig. 2). To investigate dynamic clonal evolution during ruxolitinib treatment, we performed clustering of coding mutations (synonymous and non-synonymous SNVs) from baseline and last time point WES using Sciclone[25]. Clustering of mutations by their respective copy number adjusted VAFs allowed identification of outgrowing clones over time in most patients (Supplementary Fig. 3). Looking for an enrichment of functional pathways in the genes whose mutation were lost/gained over time, we noted acquisition of mutations in genes of the RAS/RTK pathways in one third of patients (Fig. 1c). Mutations were acquired in *BRAF, CBL, KRAS, NRAS,* and *RIT1*. Of note, using pmsignature, two mutation signatures could be identified in our cohort[26]. Signature 1 showed highest similarity with COSMIC signatures 5, 19, and 30, for which the etiology remains less well understood to date. Signature 2 with a predominance of C > T transitions at CpG, a signature found in most cancer samples that correlates with age and is probably initiated by spontaneous deamination of 5-methylcytosine (Cosine similarity score = 1.0 with COSMIC signature 1 like; Fig. 1d)[27,28]. No greater signature evolution was observed during ruxolitinib treatment (Supplementary Fig. 4).

Next, we correlated genetic changes with the clinical course by comparing patients with different clinical outcomes. Two patients with the *JAK2* V617 mutation (MPN09 and MPN19) achieved a molecular remission with ruxolitinib therapy. MPN09 had a low *JAK2* V617 variant allele frequency (VAF) of 12% together with two additional mutations at low-VAF before therapy, none of which were detected at the second exome analysis (4 years later). Using ultra-deep sequencing *JAK2* V617F was detectable at very low VAFs ranging from 0.15 to 0.3% in the entire follow-up period, which was below the sensitivity threshold of exome sequencing. In MPN19, a total of 13 mutations (including *JAK2* mutation) were detected at baseline. However, strikingly in the second sample, taken three years later, a completely different set of mutations was identified and at the last time point, four years after initiation of therapy, none of the mutations were detected in the DNA sample (Supplementary Fig. 2c). To exclude the possibility of sampling mixed-up, we assessed unique germline polymorphisms at all time points, including the patients T-cells, and confirmed correct sampling.

The three patients who progressed to leukemia (MPN02 and MPN04) or to an accelerated phase (MPN18) showed a higher number of mutations compared to the other patients (mean = 19.3 +/− 7.2 vs. 11.8 +/− 4.6; *p* = 0.17 two-sided Mann–Whitney test), and all of these three cases developed mutations in *KRAS* or *NRAS* over time. As one example, MPN18 harbored hematologic cancer-associated gene mutations in *ASXL1, ETV6,* and *SRSF2* at baseline. Thereafter, additional mutations were gained in other driver genes (*IDH2* and *KRAS*). Of note, patient MPN18 acquired a second *JAK2* mutation in the kinase domain at codon R867Q (Supplementary Data 1), associated with treatment resistance to JAK inhibitors[29].

The majority of the ten patients with a durable response during JAK inhibition showed less evidence for major genetic changes with respect to the total number of mutations gained/lost or dynamic changes of allele burden (Supplementary Fig. 2b). For example, allele burden of the disease-defining *JAK2* V617F mutation and two concomitant *TET2* alterations remained stable at similarly high VAFs close to 50% over a time period of 3.5 years in MPN07. Likewise, in MPN08 *CALR* and

subclonal *TET2* mutations showed few to any changes during four years of treatment. Interestingly, in MPN11 we noted opposing dynamics of mutated *SF3B1* and *JAK2* clones; the allele burden of the *SF3B1* N626D mutation constantly decreased, whereas the *JAK2* V617F, which was initially subclonal, raised over time to become the dominant clone three years after the baseline WES, questioning a common origin of both clones. Whole-genome sequencing (WGS) provided further evidence for the independence of both clones as no shared, recently acquired somatic mutation could be identified (Supplementary Fig. 5).

Using WES data, a total of two somatic copy number alterations (CNA) were identified at baseline and nine at the last time point. The majority of patients did not show CNAs at any time point, except for those that later on transformed to leukemia (Fig. 1e). Copy-neutral loss-of-heterozygosity (CN-LOH), or uniparental disomy (UPD), affecting the *JAK2* locus at chromosome (9p24) was detected in six out of ten *JAK2*-mutated patients with apparently multiple UPDs in two of them (MPN1 and MPN10; Fig. 2b and Supplementary Figs. 6 and 7). In MPN01 these multiple UPD clones were reduced to a single-UPD clone over time by clonal selection, while a UPD affecting the *TET2* locus on chromosome (4q24) was acquired at the last time point (Supplementary Fig. 7). In addition, the existence of multiple 9pUPDs and their different clonal behavior over time, impedes tracking of *JAK2* V617F allele burden solely based on VAFs using bulk DNA (Fig. 2).

Collectively, we noted that patients with progressive disease at later time points presented with more genetic aberrations at baseline and acquired more additional aberrations over time. It is conceivable that this observation is reflecting increased genomic instability but could also be a result of stronger selective pressures. However, *JAK2* LOH does not seem to drive disease progression as all six affected patients showed long lasting responses to ruxolitinib, including one patient who achieved molecular remission.

**Allelic burden of clonal mutations in sorted cell fractions.** Next, we investigated the hematopoietic lineage repartition of gene mutations in seven ruxolitinib-treated MF patients with samples available for flow-sorting of peripheral blood (PB) cell fractions. To this aim, we first performed targeted ultra-deep resequencing of all individual patient-specific mutations identified by WES at all time points to improve the accuracy of VAF-based clone clustering methods with which we were able to reconstruct the clonal structure, as well as the dynamic evolution of predicted clusters over time[25,30]. Overall, a mean coverage of 15,250 reads/amplicon was achieved in a total of 1499 amplicons, hence allowing reliable detection of low-burden mutations (VAF ≥ 0.1% lower sensitivity limits). At least one mutation was selected as a representative of each clone cluster in addition to the disease-defining *JAK2* or *CALR* mutations. Subsequently, allelic burden was quantified in different cell fractions sorted from mononuclear cells (CD3+, CD14+, CD19+, CD34+, and CD66b+) by ultra-deep resequencing (Fig. 3 and Supplementary Figs. 8 and 9).

In all seven investigated MF patients we noted a higher allelic mutation load in the myeloid compared to the lymphoid compartment, indicating a skewed differentiation of circulating mutated CD34+ progenitors towards myeloid cell compartments in MF with only few mutations being detected at low allele frequency in B- or T-lymphocytes. In addition, three patients (MPN18, MPN01, and MPN11) had a differential mutation distribution among the different myeloid cell lineages (Fig. 3b and Supplementary Fig. 9).

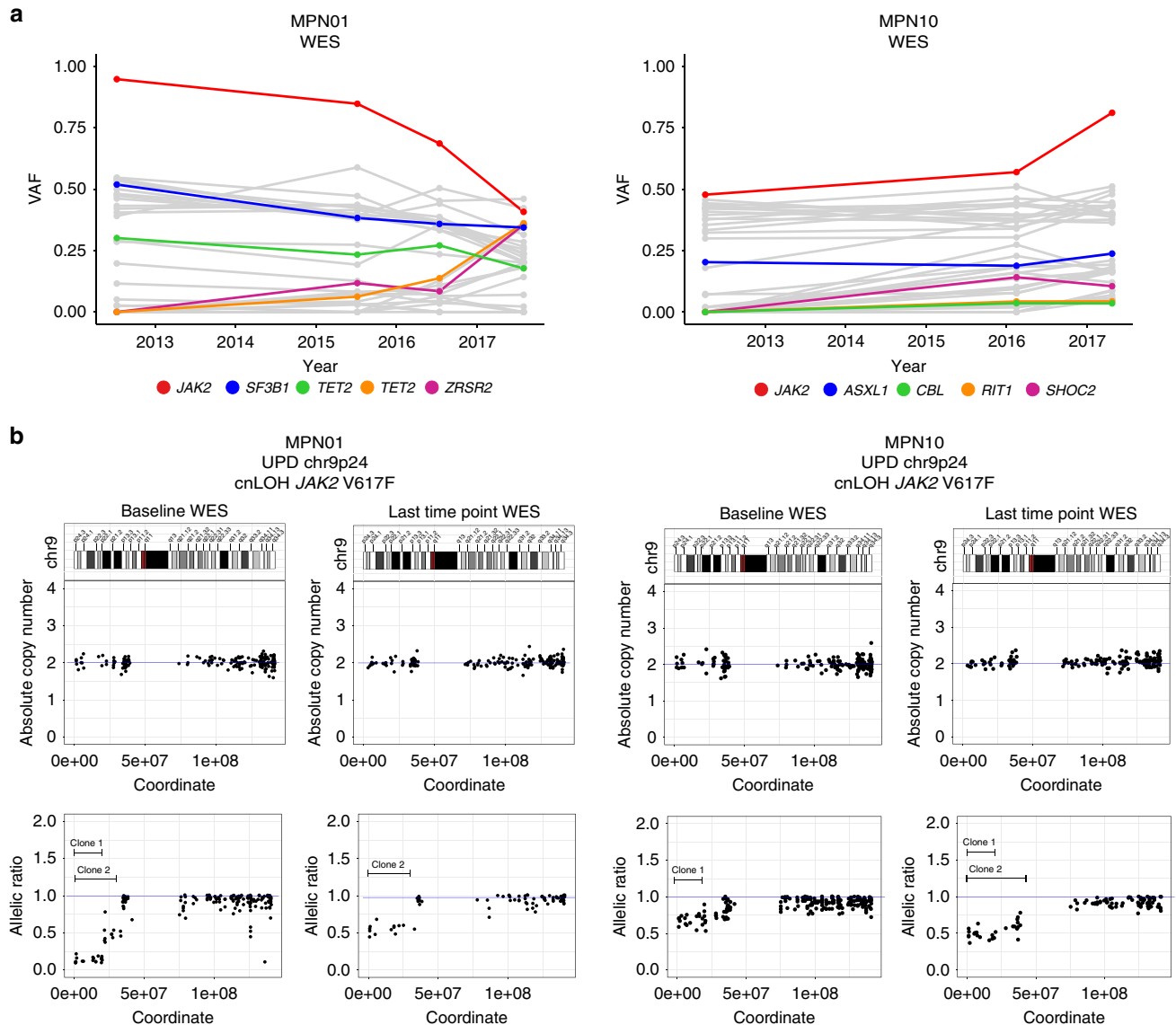

**Fig. 2 Multiple CN-LOH affecting the *JAK2* V617 locus. a** Tracking variant allele frequency by serial WES in two patients with JAK2 CN-LOH. Each patient has multiple time points analyzed (MPN01: *n* = 4; MPN10: *n* = 3) with at least 5-years of follow-up. Known driver genes with mutation are shown as colored lines, with other genes shown as gray lines. **b** Depiction of evolution of multiple chromosome 9p acquired UPDs over time by analysis of baseline and last time point WES data. Chromosome 9 ideogram with bands (top), absolute copy number (middle) and allelic ratio (bottom) values ordered by genomic coordinates. Independent clones are indicated by butted lines. Source data are provided as a Source Data file.

MPN18, who acquired several hematologic cancer-associated gene mutations over time, showed a differential segregation of mutations between monocytes and granulocytes. While all mutations were detected in the progenitor compartment, *IDH2* R140Q and *JAK2* R867Q mutations were present predominantly in granulocytes and *KRAS* G12R was mainly found in monocytes. This differential segregation of mutations between granulocytic and monocytic compartments was not restricted to the transformed MF case (MPN18), but was also present in two clinically stable patients (MPN01 and MPN11). In MPN01, a differential mutation repartition with preferential expansion of the *TET2*-mutated clone towards the monocytic population was observed (Fig. 3b), a known phenomenon described for *TET2*-mutated HSCs in chronic myelomonocytic leukemia (CMML) and clonal hematopoiesis[31,32]. Remarkably, *JAK2* V617F allelic burden was comparably low (9.1%) in monocytes compared to CD34 + progenitors (45.9%) and granulocytes (79.9%), indicating the presence of cells lacking disease-defining mutations. In the

remaining four investigated patients (MPN05, MPN10, MPN16, and MPN17), we did not observe this different segregation of mutations among the hematopoietic differentiation tree (Fig. 3b and Supplementary Fig. 9).

Collectively, these data suggest a complex clonal architecture in MF, in which evolution of clones that lack disease-defining mutations involving *JAK2* or *CALR* might be more common than expected. Thus, to better understand the clonal evolution of MF progenitors, particularly from cells lacking *JAK2* or *CALR* mutations, we next performed single-cell analysis of lineage negative (Lin-), CD34+ progenitors from 8 out of the 15 MF patients.

**Single-cell dissection of genetic architecture and phylogeny.** To genotype Lin-CD34+ progenitor cells we used a single-cell multiplexed quantitative PCR (qPCR) approach on a micro-fluidic platform (Fluidigm)[33]. Allele-specific TaqMan probes were

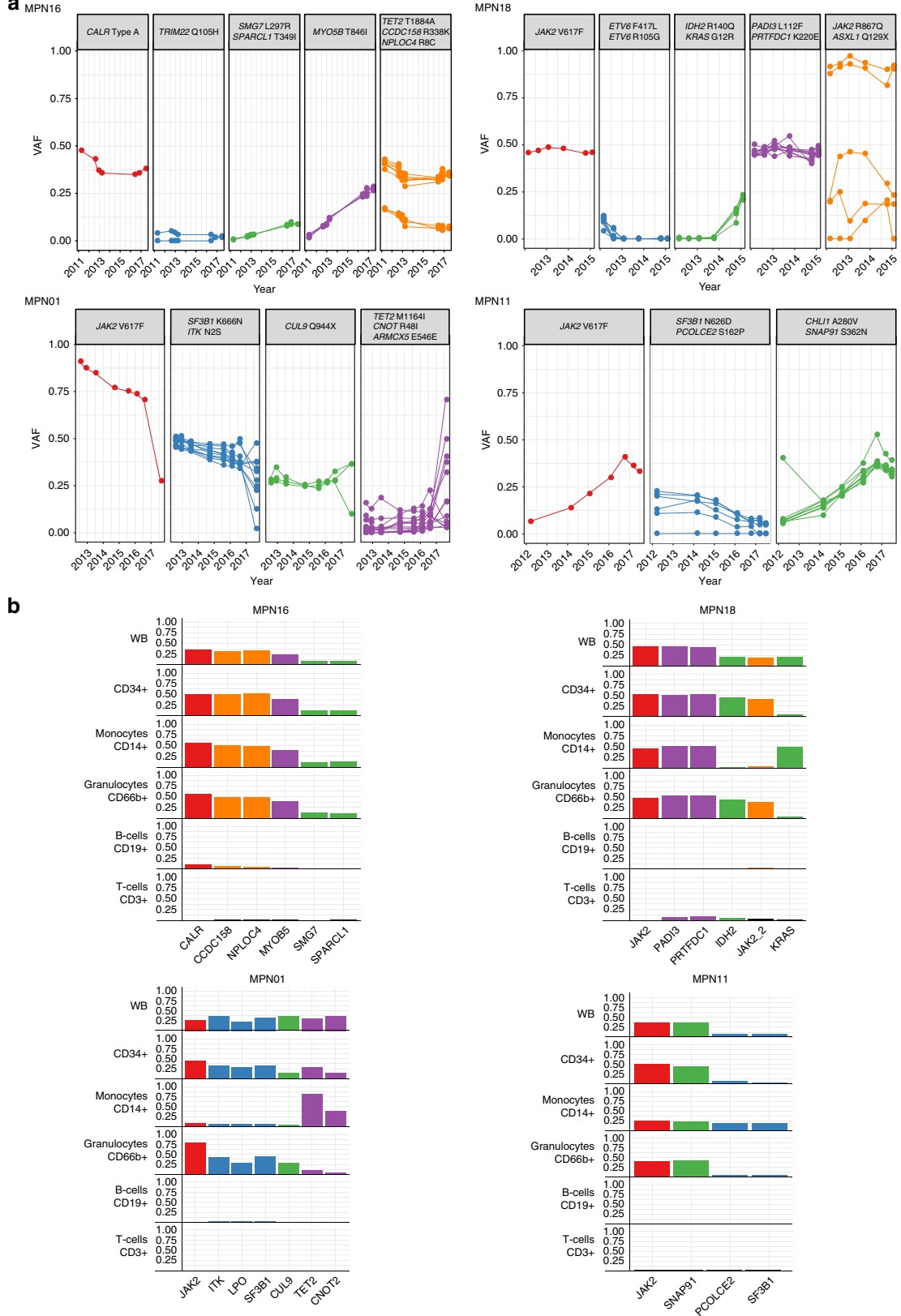

**Fig. 3 VAF-based clonal evolution analysis and allele burden quantification in flow-sorted cell fractions. a** Mutations clustered by VAF generated from ultra-deep sequencing at various follow-up time points. Disease-defining mutations in JAK2/CALR are depicted independently to emphasize their specific role in disease pathogenesis. From each cluster representative mutated genes were selected. **b** Representative mutation distribution in different blood lineages. Patients with differential segregation of mutations are displayed. Bar color correspond to respective clones shown in **a**. Source data are provided as a Source Data file.

designed for selected mutations representing VAF-predicted clones (Fig. 3a and Supplementary Fig. 8). Based on a priori power calculations for single-cell analysis, we determined that a sample size of 400 single cells can identify the presence of sub-clones at ≥2% frequency[34], this would equate to ~7 cells in our system, given an expected loss of 10% of data points. A total of 192 to 480 cells per sample were flow-sorted resulting in a total of 5184 cells across 12 time points from 8 patients that were assayed. Four patients were investigated at a second time point (MPN01, MPN04, MPN10, and MPN11) as depicted in Supplementary Fig. 1. Efficient single-cell sorting by flow cytometry was assessed by parallel plate processing of two copy-number probes (SLC2A9 and PPIP5K1 located in diploid regions of the genome) by qPCR. Sorting errors such as cell doublets or empty wells determined the mean cell sorting failure rate to be 12.5% (3.5% cell doublets and 9% empty wells per 65 attempts; Supplementary Table 4). Only cells with amplification signals within the upper and lower quartiles of respective CT values for all probes were retained for analysis. False-positive error rates (FPR) for each SNV assay were determined in K562 single-cells in a patient-specific multiplex experiment. Only 3 assays (TRPM5, SUZ12, LRCC3) generated false-positive results of ~5% of investigated K562 cells (Supplementary Table 3). These FPR were used to define a minor sub-clone threshold. Cell data from suggested subclones that did not exceed theses rates were removed. For reconstruction of the clonal phylogeny of MF and mutational co-occurrence, we were able to examine an average of six mutations (4–8 range) in 192 to 420 single-cells from patient samples. A breakdown of the single-cell data with exclusion criteria can be found in Supplementary Table 4 and on average 79% of single-cells (4113/5184 cells assayed) generated high-quality data on all interrogated mutation targets. A high correlation ($r^2 = 0.97$) was found between the allele burden detected by ultra-deep sequencing of bulk and single-cell genotyping of flow-sorted CD34 + progenitors (Supplementary Fig. 10).

Four-thousand eighty-three out of 4113 (99.3%) Lin-CD34+ progenitors were identified to harbor at least one somatic mutation. Wild-type cells were detectable only in patients MPN01 ($n = 12$) and MPN05 ($n = 18$). By manually clustering cells based on co-occurring mutations, different cell genotype groups were defined and as expected from clonal VAF models shown in Fig. 3a, JAK2 and CALR mutations were present as early mutations in all patients studied by single-cell genotyping (Fig. 4 and Supplementary Fig. 11). In most cases as primary event (MPN04, MPN05, MPN10, MPN16, MPN17, MPN18), in other cases as early secondary event (MPN01 and MPN11) in which preceding clones harboring SF3B1 K666N and SF3B1 N626D mutations were observed.

We employed a heuristic search algorithm to select a phylogenetic tree with Maximum Likelihood under a finite site model of evolution where different mutations and LOH (common events at the JAK2 locus in our cases) can occur reiteratively[35].

When looking at genetic abundances, we were able to discriminate two groups of patients, those with one dominant genotype (MPN17, MPN18, MPN04, MPN05, MPN11) and those with multiple (sub)clones of comparable clone size (MPN01, MPN10, MPN16; Fig. 4 and Supplementary Fig. 11). By detailed examination of each patient we observed interesting evolutionary events.

MPN01 presented with two independent clones: one defined by a SF3B1 K666N mutation from which the main JAK2 V617F-mutated MF subclone originated, and a second independent clone defined by a TET2 M1164I mutation. At the first time point, the JAK2 V617 clone represented 96.2% of the circulating CD34+ progenitor compartment. After 2 years of treatment with ruxolitinib, this clone decreased to 73% while in parallel the

TET2 M1164I-mutated clone expanded from 2.7% to 23.1% (Fig. 4). In parallel with the expansion of the TET2-mutated clone additional genetic events occurred within this clone, including LOH encompassing the TET2 locus on chromosome (4q24), which was also confirmed by CNA analysis using WES data. This observation might reflect positive selection of the TET2 clone and/or opportunistic expansion due to freed-clonal space by ruxolitinib treatment.

MPN04 showed a complete change in clonal architecture due to an acquired LOH of FGF1 V66M on chromosome (5q31) between the two investigated time points, before and after leukemic transformation. We noted a complete clonal sweeping by the major subclone harboring the FGF1 V66M mutation by newly emerging clone(s) that lost the mutation due to LOH and acquired additional mutations, including a NRAS G61P mutation that probably accounted for the leukemic transformation (Fig. 5). In the same patient, by single-cell genotyping we also identified a LOH at chr19p13, leading to homozygosity of CALR mutated cells from MPN04 accounting for 4% of all CD34+ progenitors at first time point. Interestingly, 19p UPDs have been reported to be more frequently found in CALR-mutated patients harboring a CALR Type-B insertion, associated with del(5q) and were more often found in accelerated disease phases[36]. All of these findings were observed in MPN04. The patient MPN16 with three subclones harboring the very same founding mutations and with a similar proportion, was assumed as a unique clone, with low diversity and with an apparent linear evolution (Supplementary Fig. 11). In the case of MPN18, two major subclones derived from an ancestral JAK2 V617F clone were observed: one at a low frequency (7.7%) defined by KRAS G12R and a second one at a high frequency (92.3%) defined by IDH2 R140Q. The second subclone subsequently acquired functionally relevant mutations, including a JAK2 R687Q mutation ("JAK2_2" in Fig. 4), which confers a resistance to JAK inhibition[29]. A clear example of convergence of an LOH of JAK2 V617F was observed in MPN17 with two major subclones derived from a JAK2-mutated founding clone: one harbored a SERPINA1 M398I and the other subclone a ARID2 R285Q mutation. Of note, 9p UPD affecting the mutated JAK2 locus occurred in both subclones independently (Fig. 4). Collectively, these data indicate that CN-LOH affecting somatically mutated drivers is a common event in MF pathogenesis that occurs not only in disease-defining mutations but also affects other genomic regions harboring rare somatically acquired mutations.

## Discussion

The tyrosine kinase inhibitor (TKI) ruxolitinib is the only targeted therapy approved for the treatment of MF. While substantial clinical benefits ameliorating MF-related symptoms and improving overall survival can be achieved[20], this non-curative therapy approach leads only to a modest decline in allele burden of disease-defining mutations in JAK2 or CALR in most patients[37]. To better understand genetic mechanisms of disease progression and resistance, we dissected clonal evolution with single-cell resolution using a combination of WES and multiplexed qPCR single-cell genotyping. Two patients achieved a molecular remission at a level of persisting residual disease of $1 \times 10^{-3}$. In one these cases (MPN19), we noted the appearance of a completely novel set of gene mutations during remission three years after initiation of ruxolitinib. A similar observation has been reported in remission samples from patients with chronic myeloid leukemia (CML) after treatment with the TKI imatinib[38]. This likely represents genetic drift during neutral evolution as a consequence of a rapid expansion after TKI. All other 13 patients showed only a modest—if any—decrease of 10–20% JAK2/CALR

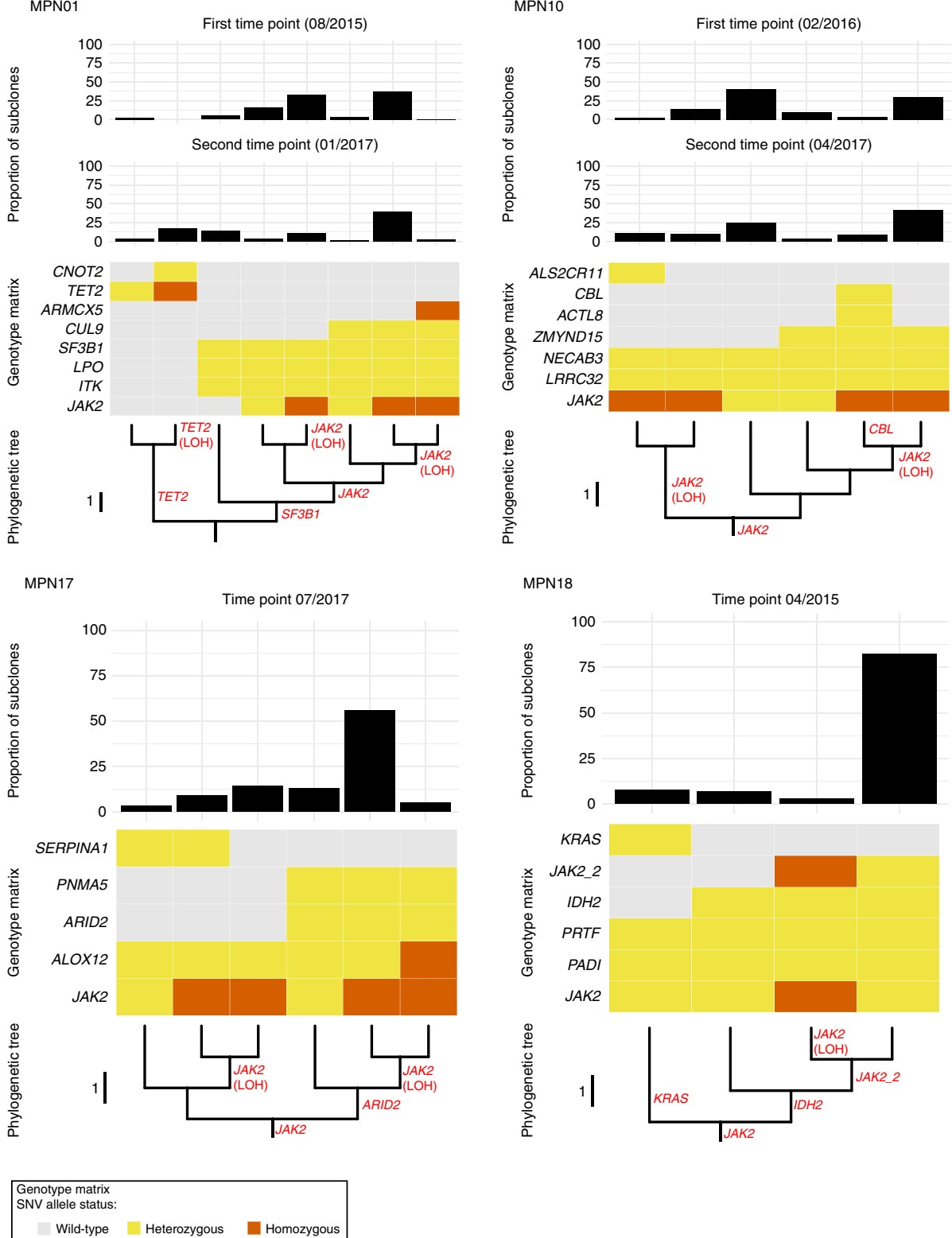

**Fig. 4 Phylogeny of CD34 + progenitors in MF and proportion of subclones.** MPN01 shows two independently originated clones, marked by a *JAK2* V617F and *TET2* M1164I mutation, respectively. Both MPN01 and MPN10 represent samples with multiple clones (cell genotypes) present with similar subclonal frequency. MPN17 and MPN18 represent samples with a dominant clone and few additional subclones. MPN01, MPN10, and MPN17 show parallel evolution of 9pUPDs (indicated by "*JAK2* (LOH)" in red text). Top panel: bar chart displaying the proportion of each observed subclone. Middle panel: genotype matrix for each subclone. Bottom panel: Evolutionary trees generated by analysis of the single-cell data. In each patient a single-phylogenetic tree was constructed and displayed as a vertically oriented rectangular cladogram. The root of the tree harbors either a JAK mutation (MPN10, MPN17, and MPN18) or a wild-type cell genotype (MPN01). Branch lengths are indicated (proportional to the number of evolutionary changes inferred) and the internal nodes (the points at which branches diverge) represent the ancestral clade from which arise all genotypes at the leaves/tips of the tree (descendant subclones). Source data are provided as a Source Data file.

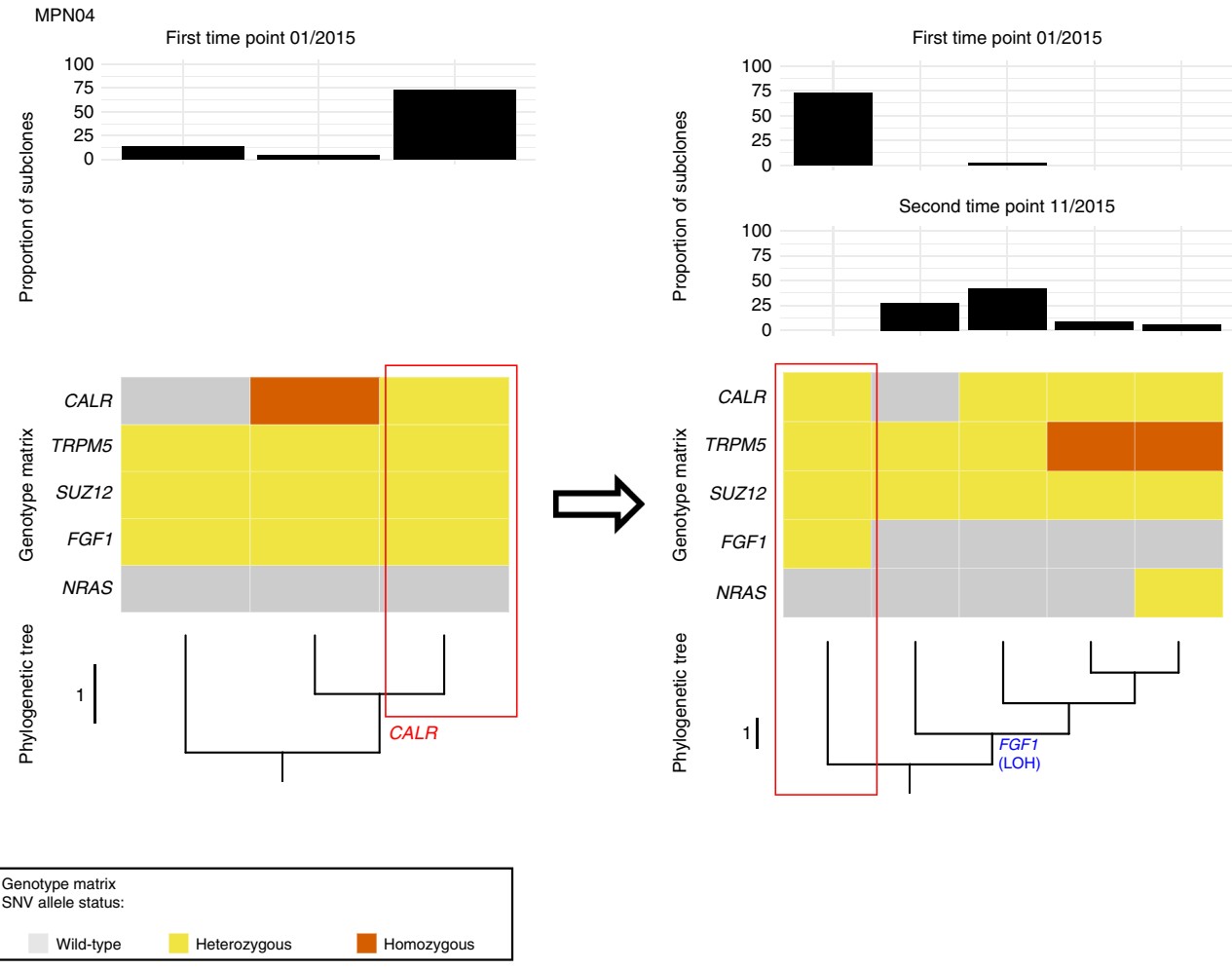

**Fig. 5 Phylogenetic Tree of CD34+ progenitors and proportion of clones in MPN04.** MPN04 showed a complete change in clonal architecture due to an acquired LOH of *FGF1* V66M on chromosome 5q31 between the two investigated time points (before and after leukemic transformation). At first time point, a dominant clone harboring mutations in *CALR, FGF1, SUZ12*, and *TRPM5* was present, from which a subclone acquired a del(5)(q23-q32), leading to wild-type *FGF1*. At the second time point, this subclone developed additional genetic abnormalities affecting *CALR* and *TRPM5*. Top panel: bar chart displaying the proportion of each observed subclone. Middle panel: genotype matrix for each subclone. Bottom panel: Evolutionary trees generated by analysis of the single-cell data. A single-phylogenetic tree was constructed and displayed as a vertically oriented rectangular cladogram. The root of the tree harbors a wild-type cell genotype. Branch lengths are indicated (proportional to the number of evolutionary changes inferred) and the internal nodes (the points at which branches diverge) represent the ancestral clade from which arise all genotypes at the leaves/ tips of the tree (descendant subclones). Source data are provided as a Source Data file.

allele burden which was often accompanied with the expansion of *JAK2/CALR*-wild-type clones due to positive selection and/or freed-clonal space under TKI treatment. In some cases, these clones showed preferential differentiation towards other myeloid cell lineages than the *JAK2* V617F-positive clones. This observation is of importance as leukemic blasts from transformed *JAK2* V617F-positive MPNs are frequently negative for the *JAK2* V617F mutation[39]. Emergence of second-site mutations in the targeted (onco-)gene is a known resistance mechanism in many myeloid malignancies and other cancers under TKI treatment[6,40]. Here, we report the acquisition of a second *JAK2* mutation at R687Q, which has been shown to confer resistance in vitro. In line with reports for other myeloid diseases, we noted frequent acquisition of mutations affecting the RAS-RTK pathways in the face of acute transformation[24,32,41]. Of particular interest, *NRAS* and *KRAS* mutations were acquired in those patients who progressed to secondary acute leukemia or accelerated phase (MPN02, MPN04, and MPN18). Of note, a recent study identified acquired mutations in *NRAS* and *KRAS* to mediate resistance to the TKI gilteritinib in *FLT3*-mutated AML[42]. This resistance could be

partially reverted by combinatorial signal inhibition using MEK inhibitors[42]. Our data show that monitoring of these mutations could provide a window for early intervention as respective clones were already present months prior to overt leukemic transformation in MF patients. Thus, a similar combinatorial approach might reflect an avenue for further exploration.

Our single-cell analysis allowed us to refine the VAF analysis-based clonal architecture, to determine the recurrence of chromosomal aberrations and to establish subclonal diversity in apparently homogeneous clones. As described, same rules of Darwinian evolution apply for tumor evolution. Therefore, phylogenetic analysis can be performed using single-cell genotyping data[3,43,44]. The monoclonal nature of MF that derives from a single stem cell harboring disease-defining *JAK2, CALR*, or *MPL* mutations has been widely studied and reviewed[45,46]. Owing to increased intrinsic proliferative activity, different clones arise due to spontaneous mutations (genetic drift) and increased genetic instability. However, a close look into subpopulations inside the MF clones identified different population dynamics. The most common clonal structure is characterized by a dominant clone which by far exceeded the

minor ones (five out of eight patients). In contrast, a relatively high richness of subclones in a similar proportion, in which no clone stands out, was observed in three out of eight patients. LOH events were found in seven out of eight patients investigated by single-cell analysis and were not restricted to the *JAK2* mutation locus. In some patients, LOH of *JAK2* V617F occurred independently in two subclones (homoplasy), a phenomenon of convergent evolution reported in other malignancies and aplastic anemia[47–49]. We also noted cases with multiple 9pUPDs, of which one got selected after ruxolitinib therapy (e.g., MPN01). LOH events gave rise to both, a mutant homozygous but also reversion to a wild-type genotype (e.g., *FGF1* in MPN04). However, also limitations of our study should be considered. Firstly, our technical approach based on mutation-specific genotyping assays does not allow consideration of false-negative genotypes as positive control samples for each mutation would be required. However, prior work using this technique showed highly reproducible data[33,50], suggesting that this weakness might have only minor effects on the conclusions drawn from these analyses. Secondly, we cannot ascribe a causative role of ruxolitinib treatment to the observed evolutionary processes due to lack of a control cohort not receiving ruxolitinib over a comparable disease time.

In summary, using the integration of serial WES, allele burden quantification in different lineages and single-cell genotyping, we created an in-depth outlook of the genetic evolution and complexity for each patient, which provided unappreciated insights into underlying genetics of MF. This approach could also be used for early detection of leukemogenic events, which can be further applied for early detection of treatment resistance or appearance of secondary diseases.

## Methods

**Sample collection and inclusion criteria.** Fifteen patients with diagnosis of primary or secondary MF were included if a PB sample at initiation of ruxolitinib treatment was available (Table 1). Thereafter, PB was sampled serially in a prospective manner (mean follow-up time: 3.9 years/patient). The study was conducted in accordance with the Declaration of Helsinki and with ethical approval obtained from the local ethics committee of the Charité—Universitätsmedizin Berlin, Germany. All patients provided written informed consent.

**Sample preparation and in vitro T-cell expansion.** Neutrophils and mononuclear cells from PB specimens were enriched by Ficoll density gradient centrifugation and were stored at –196 °C in liquid nitrogen until use. Genomic DNA was extracted from Peripheral Blood Mononuclear Cell (PBMCs) using QIAamp DNA Mini Kit (Qiagen) and sorted sub-fractions using NucleoSpin Tissue XS (Macherey-Nagel) according to the manufacturer's recommendations. For in vitro T-cells expansion $1 \times 10^6$ PBMCs were seeded on non-treated cell culture plates coated with anti-CD3 and anti-CD28 antibodies in a medium containing IL-2. Cells were split every 2 to 3 days and cultivated for a total of 10 to 12 days. Once harvested, cells were subjected to flow-sorting of CD3+T-cells.

**Whole-exome sequencing.** Whole-exome sequencing (WES) was performed in 42 samples from 15 MF patients. WES libraries were generated from whole blood DNA for tumor specimen and from in vitro expanded CD3+ T-cells for matched germline controls. Libraries were generated using 200 ng of genomic DNA using SureSelect Human All Exon V5 kit (XT protocol; Agilent). The libraries were sequenced in paired-end mode 2 × 124 bp on an Illumina Hiseq 2500 instrument (Illumina)[51,52]. Sequence alignment and mutation calling were performed using our in-house pipelines with minor modifications[51,53]. Candidate mutations with (1) Fisher's exact $p \le 10^{-1.3}$, (2) EBCall's exact $p \le 0.0001$, and (3) a VAF in tumor samples over 5% were selected. These variants were further filtered by excluding (1) synonymous SNVs, (2) SNVs in genes whose structure is not correctly annotated, and (3) SNVs listed as polymorphisms in the 1000 Genomes Project, ESP6500 and HGVD with minor allele frequency ≥ 0.001. Structural variants[54] and copy number alterations were also evaluated from WES data using our in-house pipeline CNACS[55]. CNACS is a UNIX-based program for sequencing-based copy-number analysis, which is available from web site (https://github.com/papaemmelab/toil_cnacs). For mutation signature analysis, we performed de novo extraction of signatures using pmsignature[26] for coding (synonymous and non-synonymous SNVs) and intronic mutations, which identified two signatures. Subsequently, we applied MutationalPatterns[56] to find optimal contribution of these signatures to the mutational profiles of each sample.

**Targeted sequencing.** Short fragments of 100–200 bp were PCR-amplified and pooled for library preparation. Libraries were purified, indexed (NEBNext Ultra DNA Library Prep Kit, New England Biolabs), and subsequently paired-end sequenced on a MiSeq sequencer (Illumina)[57]. Mutation-specific primers are listed in Supplementary Data 2. ddPCR was performed for *JAK2* V617F and *CALR* Type A and B variants. Assays and droplet generation were performed according to the manufacturer's guidelines. Droplets were generated on a QX200 Droplet Generator, read on a QX200 Droplet Reader and analyzed using QuantaSoft V.1.7.4 (all from Bio-Rad)[24,31].

**VAF-based clonal evolution estimation.** Clonal evolution analysis, were performed using Sciclone/ClonEvol R packages[25,30]. Data from baseline and last time point WES were used as source data, and regions with <25x depth in either of time points and ≥3 bp of indels were excluded to guarantee accurate VAFs. Tumor purities were estimated based on the allele frequencies of clonal copy number changes or VAFs of clonal driver mutations, such as mutations in *CALR* and *JAK2*. Subsequently, we estimated the cancer cell fraction (CCF) for each mutation based on purity (p), local tumor copy number (CNt), local normal copy number (CNn) according to the following formula: $\mathrm{CCF} = \mathrm{VAF} \times (1/p) \times (p \times \mathrm{CNt} + \mathrm{CNn}(1 - p))$. Copy number adjusted VAFs were obtained by dividing above CCFs by 2, which were used in the input of Sciclone[58]. Results were visualized as plots depicting the clonal dynamics of expanding and vanishing mutations on time (Supplementary Fig. 8).

**Cell flow-sorting.** For sorting of lineage negative(Lin-) CD34+ cells, PBMCs cells were thawed at 37 °C and cultured overnight in RPMI medium, supplemented with 10% FBS and 1x Streptavidin and 1x Penicilin/Streptomycin at 5% $CO_2$, 37 °C. Cells were washed and lineage-positive cells were labeled using Human Cell Depletion Set (BD Biosciences). Lin+ cells were depleted using streptavidin coated magnetic beads. The supernatant containing Lin− cells was washed and labeled with anti-human CD34-PE antibodies (BD Biosciences). Remaining Lin+ cells were labeled with Streptavidin-BrilliantViolet (BD Biosciences) conjugate for cytometric exclusion. Lin-/CD34+ single-cells were sorted in 96 well plates containing 2.5 μL of lysis buffer. All the procedures were performed at 4 °C until completion of cell lysis. Gating procedures are depicted in the Supplementary Fig. 12.

For sorting of mature blood cell lineages, PBMCs were thawed and cultured overnight at standard conditions. Cells were then separated using the first fraction for Lin-/CD34+ sorting and the second fraction for sorting mature cell populations. For the later, cells were labeled with the following conjugated anti-human antibodies: FITC-CD3, APC-CD14, PE/Cy7-CD19 (BD Biosciences). If available, previously Ficoll-enriched granulocytes were thawed and labeled with anti-human CD66b-PE (BD Biosciences). Sorted fractions were then used for DNA extraction (Supplementary Fig. 12). Antibodies and their respective dilutions are listed in Supplementary Table 1.

**Single-cell genotyping.** Single-cell genotyping was performed following the procedure previously described[2] Briefly, allele-specific TaqMan probes (Thermo-Fisher) were designed for selected mutations representing the previously calculated clusters (VAF-based clonal evolution)[33], probes labeled with VIC detected WT allele, and probes labeled with FAM detected mutant allele in every case. For *JAK2* V617F, *IDH2* R140Q, *KRAS* G12R, *NRAS* Q61P, commercially available probes were used (ThermoFisher). In Supplementary Tables 2–4 and Supplementary Data 3 a summary of designed probes, sequences, and quality control assessments are outlined. Lin-CD34 + cells were single-cell sorted in 96 well plates, lysed and DNA was pre-amplified with the multiplexed-specific TaqMan probes and the TaqMan Preamplification Master Mix from ThermoFisher. Pre-amplified material was used for high-throughput qPCR reactions carried on a BioMark HD using 192.24 dynamic array plates (Fluidigm). Data was collected and images were inspected manually.

For each signal, manual inspection of amplification curves and amplification threshold setting was done. Reactions under the threshold were considered to be negative. Reactions with negative values for both probes, were considered failed and cells that presented at least one failed reaction were discarded. Wells with no reaction were considered empty. A summary of number of cells processed and the total number of cells used for each patient and experiment is shown in Supplementary Table 4.

For each sample, an additional calibration plate was sorted in parallel for the estimation of sorting errors as shown in Supplementary Fig. 13. Copy Number TaqMan probes were used for the estimation of doublets rates (Supplementary Table 2). False-discovery rates were determined using the K562 cell line, and estimating proportion of false-positive single-cells per probe (Supplementary Table 3). Plate processing was carried out simultaneously for each sample. An overview of the entire experimental design and procedure is shown in Supplementary Fig. 14.

*CALR* mutations, *IDH2* R140Q, and *ARMCX5* E546E could not be detected in the Biomark HD system. In the case of *CALR* mutation type B, *IDH2* R140 and *ARMCX5* E546E, standard qPCR reactions were performed (Applied Biosystems) and for *CALR* mutation type A, a ddPCR was used (Bio-Rad). Reaction data from passing cell reactions were coded as follow, 0: WT, 1: heterozygous, 2: LOH

(homozygous for the mutant). Data matrices were the used as input for the estimation of the clonal composition and phylogenetic analysis.

**Phylogenetic analysis**. Based on the presence of mutations, cells were grouped into clusters, that we refer as genotypes (Supplementary Fig. 9). Proportion of each genotype was therefore calculated (Fig. 4), and those genotypes (low-frequency subclonal cell populations) represented with a proportion of 2% or less, were considered technical errors below our FDR cut-off (FDR = 2%; Supplementary Tables 3 and 4), and removed from analysis. A 2% FDR in our experiments is equivalent to a minimum of four single cells for 192 cells profiled (MPN11_t1), eight single cells for 384 cells profiled (MPN05, MPN17, and MPN18), or ten single-cells for 480 cells profiled (MPN01, MPN04, MPN10, MPN11_t2, and MPN16). As each assay varied slightly in error rate (Supplementary Table 3) we paid additional consideration, using the following criteria to define a "bona-fide" cell population at low frequency by: (i) Independent manual review of the data matrix prior to phylogenetic analysis by a second investigator who did not perform the single-cell genotyping. (ii) An observed genotype must be attributed to four or more cells (our 2% FDR hard cut-off, dependent on the number of cells profiled for each patient; See above). (iii) A single-SNV/LOH event cannot define a low frequency/minor subclonal population if the population is less than the error rate for that given SNV assay as described in Supplementary Table 3.

For Maximum Likelihood/Finite sites assumptions, heuristic searches of Maximum Likelihood trees, assuming finite sites assumption was performed using SiFit[35], for each sample, ternary genotype matrix was used. For tree visualizations and plotting the R package ggtree was used.

**Reporting summary**. Further information on research design is available in the Nature Research Reporting Summary linked to this article.

## Data availability

The authors declare that the data supporting the findings of this study are available within the paper and its extended data files. All baseline and last time point WES data have been uploaded on EGA (Accession ID: EGAS00001003829). Source data underlying Figs. 1–5 and Supplementary Figs. 1–11 and 13 are provided as a Source Data file. All other data are available from the corresponding author upon reasonable requests.

## Code availability

No new code was produced for the analysis of the data in this manuscript. Details of the computer code used are included in the Methods section.

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

## Acknowledgements

This study was supported by a grant from the Boehringer Ingelheim Stiftung, grant 18/6 from the Gutermuth Stiftung, and grant #2017_EKES.33 from the Else Kröner-Fresenius-Stiftung all awarded to F.D., a DKTK research grant awarded to F.D. and L.B, and grant #FRFF201625 from the Berliner Krebsgesellschaft awarded to M.F.; M.F. and D.N. were supported by the BIH Clinician Scientist Fellowship Program. W.C. received a fellowship from the Deutsche José Carreras Leukämie-Stiftung. M.J.J.R.-Z. received a John Goldman fellowship from the Leuka Charity (2016/JGF/0003). This work was partially sponsored by Project for Cancer Research and Therapeutics Evolution (P-CREATE) from Japan Agency for Medical Research and Development (16cm0106501h0001) to S.O. We would like to thank Antje Maluck, Helga Fleischer-Notter, Katrin Vogt, Irina Lehmann, and Loreen Thürman for technical assistance and/or access to the Fluidigm technology. We would like to acknowledge the assistance of the BCRT Flow Cytometry Lab.

## Author contributions

E.M. and F.D. designed the research; E.M., M.F., K.Y., F.C., K.H., M.O., D.N., C.H., W.C., Y.O., Y.Shir., Y.Shio., T.Z., C.C.O., B.S., M.J.J.R-Z. and S.O. performed the research and/or bioinformatics. M.F., J.K., M.S., L.B. and P.C. contributed patient samples and clinical data; E.M., M.F., K.Y., C.C.O., S.O., and F.D. analyzed the data; E.M., K.Y., L.B., M.J.J.R-Z, S.O. and F.D. wrote the paper. All authors read and agreed to the final version of the manuscript.

## Competing interests

The authors declare the following competing interests: F.D. received research funding from Novartis. P.C. received speakers honoraria from Novartis, BM, Pfizer, and Incyte. All other authors declare no competing interests.
