## [Peer Review File · Nature Communications]

Reviewers' comments:

Reviewer #1 (Remarks to the Author): Expertise in single-cell analysis

Mylonas, Yoshida & Frick et al describe clonal evolution patterns in myelofibrosis patients during the course of treatment with JAK inhibitors. Starting with a foundation of whole exome sequencing performed between 2 and 4 times during the course of treatment, the authors catalog detected mutations and trace their prevalence over time. To supplement their exome data, for a subset of their patients and samples they performed targeted deep sequencing on myeloid and lymphoid cell fractions. Finally, for a further subset of mutations, they used a single cell multiplex qPCR assay to obtain a digital readout of their measures of VAF previously obtained by WES and targeted sequencing.

The manuscript is a summary of a large and valuable collection of longitudinal genomic data and is mostly descriptive with limited mechanistic insights - somewhat understandably limited by the nature of the samples under study. The clonal histories of the tumors in each patient are described and while the authors describe themes linking different patients surrounding their heterogeneity and the decline or increase in allelic fractions, the study is underpowered to make conclusions so these themes are really just a collection of interesting observations.

Specific Points:

Figure 3a - no gene name labels on the 2nd panel.

Some of the language seems overly strong: mutations being "only" present in 1 compartment or "exclusive" to another. The plot is small and hard to read, but there is clearly non-zero levels of KRAS in granulocytes and JAK2 in B cells and monocytes

Figure 4. Were false negatives considered? In many cases, the most parsimonious explanation for the presented clonal relationships would be the failure to detect the mutation of interest because of technical reasons.

In general it is hard to ascribe a causative role for ruxolitinib to the evolutionary processes shown because of the absence of controls that were not treated with the drug (or treated with another drug). Clearly there are ethical reasons why this is not the case, but conclusions should be appropriately careful.

Reviewer #2 (Remarks to the Author): Expertise in MF and clonal evolution

This is a well performed study that adds evidence to the dynamic nature of subclonal evolution in MF. The number of patients is relatively small but the longitudinal sampling, the sorted populations, and the single cell analyses are a plus.

On general question is about the source of DNA. Is PBMNC OK? Has anyone ever tested its sensitivity as compared to BM analysis?

Specific questions:

Figure 1

Clonal evolution is reduced to a binary state of mutation present-absent. However, a more sophisticated analysis would be required, including estimation of the sample ploidy and purity, and comparison of the cancer cell fraction between serial samples to also ascertain the relative enrichment or depletion of subclones. Here, silent mutations would also help adding increased resolution to the analysis

The analysis of signatures is quite shallow and poorly described. First, signature extraction is apparently performed on non-synonymous variants only, which is sub-standard. Mutational processes are as likely to produce syn than non-syn mutations, and the former should be included in such analysis to increase its accuracy. Likely, even intronic mutations

flanking exons could be introduced here.

Next, the signature 1 extracted is likely a dirty sig 5-sig 8. Repeated analysis with higher number of variants, and perhaps using a de novo extraction approach followed by a second fitting approach restricted to the shortlist of identified signatures could return a cleaner result (see for example the package `mutationalpatterns` in `bioconductor`)

Figure 2

2 CNAs per patient at baseline is in line with what expected or could it be a dilution effect from PB sampling?

Figure 2c does not show increased genomic complexity at progression, or at least it does not do it in a visually attractive manner. It is a show of individual patient examples. It would be nice to have a figure comparing progressive and non-progressive cases and showing differences in overall mutation number, CNAs, evolution, etc

Figure 4:

This is the main strength of the paper and it would deserve a better representation. The diagrams shown are quite hard to figure out. Can the authors draw an actual phylogenetic tree for each patient, annotating main genomic events driving the various branches? This would also help figuring out the timing of occurrence of the subclones, something not readily visible here.

I also believe that MPN04 should deserve a place in main Fig 4.

Response to reviewers

NCOMMS-19-20549 "Single-cell analysis based dissection of clonality in myelofibrosis"

We thank our reviewers for their helpful and insightful comments. We believe that we have now fully addressed each point by providing additional information and performing further analyses. Below, we provide point-by-point responses to all comments and refer our reviewers to the relevant parts of the manuscript addressing their comments. For convenience, we use red font to refer to manuscript changes/additions in the responses below and do the same to highlight text changes in the manuscript itself. The reviewers' comments are included below in non-bold typeface and our responses are highlighted in bold.

Reviewer #1 (Remarks to the Author): Expertise in single-cell analysis

Mylonas, Yoshida & Frick et al describe clonal evolution patterns in myelofibrosis patients during the course of treatment with JAK inhibitors. Starting with a foundation of whole exome sequencing performed between 2 and 4 times during the course of treatment, the authors catalog detected mutations and trace their prevalence over time. To supplement their exome data, for a subset of their patients and samples they performed targeted deep sequencing on myeloid and lymphoid cell fractions. Finally, for a further subset of mutations, they used a single cell multiplex qPCR assay to obtain a digital readout of their measures of VAF previously obtained by WES and targeted sequencing.

The manuscript is a summary of a large and valuable collection of longitudinal genomic data and is mostly descriptive with limited mechanistic insights - somewhat understandably limited by the nature of the samples under study. The clonal histories of the tumors in each patient are described and while the authors describe themes linking different patients surrounding their heterogeneity and the decline or increase in allelic fractions, the study is underpowered to make conclusions so these themes are really just a collection of interesting observations.

Specific Points:

Figure 3a - no gene name labels on the 2nd panel.

⇒ **We now added gene symbol and mutation nomenclature labels to the 2nd panel.**

Some of the language seems overly strong: mutations being "only" present in 1 compartment or "exclusive" to another. The plot is small and hard to read, but there is clearly non-zero levels of KRAS in granulocytes and JAK2 in B cells and monocytes

⇒ We thank the reviewer for this cautionary note and adopted our wording accordingly. Replacing the words 'exclusively' and 'only' with 'predominantly' and 'mainly', respectively. The reviewer is absolutely right that *KRAS* burden is non-zero in other cell fractions (e.g. actual VAF in granulocytes: 3.8%). All allelic burden quantifications from each patient and respective subpopulations are now shown accompanying Source Data file for Figure 3 and Supplementary Figure 9.

Figure 4. Were false negatives considered? In many cases, the most parsimonious explanation for the presented clonal relationships would be the failure to detect the mutation of interest because of technical reasons.

⇒ Again, we thank the reviewer to point to this important issue. As each mutation/CNV assay is patient-specific and no single positive control sample is available, we mitigated the impact of false negative errors in our data accordingly:

(1) TaqMan-based allelic discrimination assays were used to detect SNVs, where each assay generates a fluorescent signal either for the wild-type sequence (VIC dye), the mutant sequence (FAM dye) or both (VIC and FAM fluorescence; the most common single-cell genotype in our study). Our control cells (K562 cell line; ran during our assay optimization experiments) produced robust wild-type and mutant signals in all cells profiled confirming the assay efficiency in the multiplex pre-amplification and qPCR system. Our false positive error rates for SNVs assays are described in Supplemental Table 3.

(2) We were also careful to compare the mutation allele burdens estimate generated by each genomic approach in this study (Supplementary Figure 10). These data reveal a high concordance between ultra-deep sequencing and single-cell genotyping ($r^2 = 0.97$) results.

(3) For our phylogenetic analyses we used our assay error rates to define and retain a low frequency/ minor sub-clonal population. Single-cells were removed from the analysis, included those wells that showed no or low-quality data. Data from suggested minor sub-clonal populations that did not exceed assay error rates were also removed as stated in the Phylogenetic Analysis: Clonal composition section of the methods: 'Proportion of each genotype was therefore calculated (Fig. 4), and those taxa represented with a proportion of 2% or less, were considered technical errors (FDR=2%), and removed from analysis.' We have now updated this to read: **Proportion of each genotype was therefore calculated (Fig. 4), and those genotypes (low-frequency sub-clonal cell populations) represented with a proportion of 2% or less, were considered technical errors below our FDR cut-off (FDR=2%; Supplementary Tables 3-4), and removed from analysis. A 2% FDR in our experiments is equivalent to a minimum of 4 single cells for 192 cells profiled (MPN11_t1), 8 single cells for 384 cells profiled (MPN05, 17, and 18) or 10 single-cells for 480 cells profiled (MPN01, 04, 10, 11_t2, and 16).**

(4) Furthermore, to provide additional clarity to the reader we have included the following statement in the methods, Phylogenetic Analysis: Clonal composition section after the modified statement above in (3): **'As each assay varied slightly in error rate (Supplementary Table 3) we paid additional consideration, using the following criteria to define a 'bona-fide' cell population at low frequency by: (i) Independent manual review of the data matrix prior to phylogenetic analysis by a second investigator who did not perform the single-cell genotyping. (ii) An observed genotype must be attributed to four or more cells (our 2% FDR hard cut-off, dependent on the number of cells profiled for each patient; See above). (iii) A single SNV/LOH event cannot define a low frequency/ minor sub-clonal population if the population is less than the error rate for that given SNV assay as described in Supplementary Table 3.'**

(5) We added a cautionary note indicating that false negatives could not be considered in this experimental work-up and might have slightly influenced our analysis to the discussion: **However, also limitations of our study should be considered. First, our technical approach based on mutation-specific genotyping assays does not allow consideration of false negative genotypes as positive control samples for each mutation would be required. However, prior work using this technique showed highly reproducible data,(33, 51) suggesting that this weakness might have only minor effects on the conclusions drawn from these analyses.**

In general, it is hard to ascribe a causative role for ruxolitinib to the evolutionary processes shown because of the absence of controls that were not treated with the drug (or treated with another drug). Clearly there are ethical reasons why this is not the case, but conclusions should be appropriately careful.

⇒ **We fully agree that we cannot provide evidence for a causative role for ruxolitinib to the observed evolutionary processes. At the beginning of the project we tried to collect serial samples from MF patients not receiving JAK inhibition. However, this obviously failed as ruxolitinib is nowadays standard care for MF patients with splenomegaly and/or other disease-related symptoms. We added such a cautionary note to the discussion: **Secondly, we cannot ascribe a causative role of ruxolitinib treatment to the observed evolutionary processes due to lack of a control cohort not receiving ruxolitinib over a comparable disease time.****

Reviewer #2 (Remarks to the Author): Expertise in MF and clonal evolution

This is a well performed study that adds evidence to the dynamic nature of subclonal evolution in MF. The number of patients is relatively small but the longitudinal sampling, the sorted populations, and the single cell analyses are a plus.

One general question is about the source of DNA. Is PBMNC OK? Has anyone ever tested its sensitivity as compared to BM analysis?

⇒ **We thank the reviewer for this question. Of course, we agree that BM should be the preferred source of research material in almost all hematologic malignancies. Myelofibrosis, however, should be considered an exception. Due to fibrotic remodeling of the bone marrow, regular hematopoiesis is more or less repressed, leading to extramedullary hematopoiesis in liver and spleen. For the great majority of patients, BM aspirates are impossible to obtain due to fibrosis (so called dry tap aspiration). Therefore, peripheral blood, PBMCs, isolated granulocytes or a mixture of all three sources were used instead of BM in numerous large studies published in high-ranked journals, highlighting the fact that the use of PB cells is widely accepted – and without practical alternative – in the research of myelofibrosis:**

- **Grienfeld et al., Classification and Personalized Prognosis in Myeloproliferative Neoplasms, NEJM, 2018**
- **Pacilli et al., Mutation landscape in patients with myelofibrosis receiving ruxolitinib or hydroxyurea, Blood Cancer Journal, 2018**
- **Nangalia et al., Somatic CALR Mutations in Myeloproliferative Neoplasm with unmutated JAK2, NEJM, 2013**

The use of PB instead of BM is further supported by a study published by Takahashi et al., Blood, 2013, which compared JAK2 V617F mutation burden in PB and BM. Interestingly, the sensitivity and specificity of detecting JAK2 p.V617F in PB were both 100% compared with BM. Additionally, the JAK2 p.V617F allele burden measured in PB was equivalent to that in BM ($R^2 = 0.991$; $P < .0001$).

Specific questions:

Figure 1

Clonal evolution is reduced to a binary state of mutation present-absent. However, a more sophisticated analysis would be required, including estimation of the sample ploidy and purity, and comparison of the cancer cell fraction between serial samples to also ascertain the relative enrichment or depletion of subclones. Here, silent mutations would also help adding increased resolution to the analysis.

=> We fully agree that clonal evolution is more than a binary state of present-absent but rather a dynamic process as also shown in former Supplementary Fig. 2 by serial WES throughout the disease course. In the revised version, we now added

Sciclone analysis comparing baseline and last time point WES. All coding mutations (including silent SNVs) were used and clustered according to their respective copy number adjusted variant allele frequencies (aVAFs). Of note, patients MPN09 and MPN19 with molecular remission were excluded from this Sciclone analysis due to low cancer cell fractions in the WES data from the last time point following Ruxolitinib treatment. Sciclone analysis requires two serial datasets to robustly infer the subclonal architecture of the tumors. The respective analysis is mentioned in the first results chapter: **To investigate dynamic clonal evolution during ruxolitinib treatment, we performed clustering of coding mutations (synonymous and non-synonymous SNVs) from baseline and last time point WES using Sciclone.²⁵ Clustering of mutations by their respective copy number adjusted VAFs allowed identification of outgrowing clones over time in most patients (Supplementary Fig. 3).**

The analysis of signatures is quite shallow and poorly described. First, signature extraction is apparently performed on non-synonymous variants only, which is sub-standard. Mutational processes are as likely to produce syn than non-syn mutations, and the former should be included in such analysis to increase its accuracy. Likely, even intronic mutations flanking exons could be introduced here. Next, the signature 1 extracted is likely a dirty sig 5-sig 8. Repeated analysis with higher number of variants, and perhaps using a de novo extraction approach followed by a second fitting approach restricted to the shortlist of identified signatures could return a cleaner result (see for example the package `mutationalpatterns` in bioconductor)

⇒ **We thank the reviewer for the critical evaluation of our data and apologize for our mistake. In the revised version, we now consider all coding and intronic SNVs which increased the number of mutations from 338 to 815. As shown in the new Figure 1d, cosine similarity of COSMIC signature 5 and Sig_1 slightly increased from 0.79 to 0.83. In addition, we also performed a second fitting using `MutationalPatterns` as suggested. We now added a more detailed description of our analysis in the Material&Methods section “whole-exome sequencing”: **For mutation signature analysis, we performed de novo extraction of signatures using `pmsignature`²⁶ for coding (synonymous and non-synonymous SNVs) and intronic mutations, which identified two signatures. Subsequently, we applied `MutationalPatterns`⁵⁸ to find optimal contribution of these signatures to the mutational profiles of each sample.****

Figure 2

2 CNAs per patient at baseline is in line with what expected or could it be a dilution effect from PB sampling?

⇒ **As already indicated in detail in our answer to your first question, peripheral blood and bone marrow seem to be equivalent in terms of sensitivity and specificity in the genetic analysis of myelofibrosis. Therefore, we think that a dilution effect is unlikely. This notion is supported by a large study by Grienfeld *et al.*, NEJM, 2018, investigating genetic aberrations in myeloproliferative neoplasms (including the less aggressive diseases ET and PV). There, the investigators report chromosomal gains and losses in 5-6% of all patients. Likewise, we note CNAs at baseline in a**

minority of patients (2/15 = 13%). Please note, that CNAs are now depicted in Figure 1e of the revised manuscript

Figure 2c does not show increased genomic complexity at progression, or at least it does not do it in a visually attractive manner. It is a show of individual patient examples. It would be nice to have a figure comparing progressive and non-progressive cases and showing differences in overall mutation number, CNAs, evolution, etc

We agree that that Figure 2 appeared as a collection of individual patient examples. Therefore, we decided to move former Figure 2a to revised Figure 1e. In addition, we now focus on MPN1 and MPN10 in which evidence of multiple UPDs affecting the JAK2 locus could be identified by WES. We nicely show that these clones can be differently selected (MPN01) but also acquired over time (MPN10). This fact makes simple molecular tracking of the JAK2 allele burden much more complicated. In the results chapter “whole-exome sequencing of samples during ruxolitinib therapy”, it is now written: **In addition, the existence of multiple 9p UPDs and their different clonal behavior over time, impedes tracking of JAK2 V617F allele burden solely based on VAFs using bulk DNA (Fig. 2).**

Figure 4:

This is the main strength of the paper and it would deserve a better representation. The diagrams shown are quite hard to figure out. Can the authors draw an actual phylogenetic tree for each patient, annotating main genomic events driving the various branches? This would also help figuring out the timing of occurrence of the subclones, something not readily visible here.

⇒ **We thank the reviewer for appreciating our single-cell approach. In the revised version of the manuscript, we now added main genomic events at respective branches. Additionally, the output has undergone some stylistic and annotation improvements that collectively should help to better understand these phylogenetic trees. Please see the revised Figures 4 and 5.**

I also believe that MPN04 should deserve a place in main Fig 4.

⇒ **We thank the reviewer for pointing to MPN04 and agree that this patient should be shown in the main manuscript. MPN04 is now shown in main Figure 5.**

REVIEWERS' COMMENTS:

Reviewer #1 (Remarks to the Author):

The authors have gone to considerable lengths to satisfactorily provide clarifications to the points raised in my review. I thank them for their efforts and clear explanations.

Reviewer #2 (Remarks to the Author):

The authors have adequately addressed the points raised